# Serum Proteomic Analysis for New Types of Long-Term Persistent COVID-19 Patients in Wuhan

Cuidan Li,[a] Liya Yue,[a] Yingjiao Ju,[a,b] Jie Wang,[a,b] Mengfan Chen,[a,b] Hao Lu,[a,b] Sitong Liu,[a,b] Tao Liu,[a,b] Jing Wang,[c] Xin Hu,[c] Bahetibieke Tuohetaerbaike,[c] Hao Wen,[c] Wenbao Zhang,[c] Sihong Xu,[d] Chunlai Jiang,[e] Fei Chen[a,b,c,f]

[a]Beijing Institute of Genomics, Chinese Academy of Sciences, China National Center for Bioinformation, Beijing, China
[b]University of Chinese Academy of Sciences, Beijing, China
[c]State Key Laboratory of Pathogenesis, Prevention and Treatment of High Incidence Diseases in Central Asia, Urumqi, Xinjiang, China
[d]Division II of *In Vitro* Diagnostics for Infectious Diseases, Institute for *In Vitro* Diagnostics Control, National Institutes for Food and Drug Control, Beijing, China
[e]National Engineering Laboratory for AIDS Vaccine, School of Life Science, Jilin University, Changchun, China
[f]Beijing Key Laboratory of Genome and Precision Medicine Technologies, Beijing, China

Cuidan Li, Liya Yue, and Yingjiao Ju contributed equally to this work. Authors order was determined based on more devotion to manuscript writing and/or data analysis.

**ABSTRACT** The emergence of a new type of COVID-19 patients, who were retested positive after hospital discharge with long-term persistent SARS-CoV-2 infection but without COVID-19 clinical symptoms (hereinafter, LTPPs), poses novel challenges to COVID-19 treatment and prevention. Why was there such a contradictory phenomenon in LTPPs? To explore the mechanism underlying this phenomenon, we performed quantitative proteomic analyses using the sera of 12 LTPPs (Wuhan Pulmonary Hospital), with the longest carrying history of 132 days, and mainly focused on 7 LTPPs without hypertension (LTPPs-NH). The results showed differential serum protein profiles between LTPPs/LTPPs-NH and health controls. Further analysis identified 174 differentially-expressed-proteins (DEPs) for LTPPs, and 165 DEPs for LTPPs-NH, most of which were shared. GO and KEGG analyses for these DEPs revealed significant enrichment of "coagulation" and "immune response" in both LTPPs and LTPPs-NH. A unity of contradictory genotypes in the 2 aspects were then observed: some DEPs showed the same dysregulated expressed trend as that previously reported for patients in the acute phase of COVID-19, which might be caused by long-term stimulation of persistent SARS-CoV-2 infection in LTPPs, further preventing them from complete elimination; in contrast, some DEPs showed the opposite expression trend in expression, so as to retain control of COVID-19 clinical symptoms in LTPPs. Overall, the contrary effects of these DEPs worked together to maintain the balance of LTPPs, further endowing their contradictory steady-state with long-term persistent SARS-CoV-2 infection but without symptoms. Additionally, our study revealed some potential therapeutic targets of COVID-19. Further studies on these are warranted.

**IMPORTANCE** This study reported a new type of COVID-19 patients and explored the underlying molecular mechanism by quantitative proteomic analyses. DEPs were significantly enriched in "coagulation" and "immune response". Importantly, we identified 7 "coagulation system"- and 9 "immune response"-related DEPs, the expression levels of which were consistent with those previously reported for patients in the acute phase of COVID-19, which appeared to play a role in avoiding the complete elimination of SARS-CoV-2 in LTPPs. On the contrary, 6 "coagulation system"- and 5 "immune response"-related DEPs showed the opposite trend in expression. The 11 inconsistent serum proteins seem to play a key role in the fight against long-term persistent SARS-CoV-2 infection, further retaining control of COVID-19 clinical symptom of LTPPs. The 26 proteins can serve as potential therapeutic targets and are thus valuable for the treatment of LTPPs; further studies on them are warranted.

Address correspondence to Fei Chen, chenfei@big.ac.cn.

The authors declare no conflict of interest.

**KEYWORDS** COVID-19, SARS-CoV-2, retest positive, long-term persistent COVID-19 patients (LTPPs), coagulation

Coronavirus disease 2019 (COVID-19), caused by severe acute respiratory syndrome coronavirus 2 (SARS-CoV-2), is a global pandemic disease, posing unprecedented threats to the public health (1–4). It is generally divided into severe, mild, moderate, and asymptomatic cases based on differentially clinical symptoms (https://www.covid19treatmentguidelines.nih.gov/overview/clinical-spectrum/). Recent studies have reported the emergence of a new type of long-term persistent COVID-19 patients (LTPPs), who were retested positive after hospital discharge (with long-term persistent SARS-CoV-2 infection but without any clinical symptoms), presenting new challenges to COVID-19 treatment and prevention (5–10).

To date, research about LTPPs mainly focused on case reports, risk factors, and virus genome analysis (5–10). Many case reports have indicated that approximately 15% (14%–16.7%) discharged patients were tested positive even after recovering from COVID-19, and as they are asymptomatic, they can be considered to be LTPPs (11–16). Further, some studies have analyzed pertinent risk factors, some of which include lack of antibodies binding to sequential epitopes of SARS-CoV-2, lower titers of neutralizing antibodies, and immunosuppression (17–20). In addition, the intestine as the "reservoir of SARS-CoV-2" might be one risk factor of LTPPs even if SARS-CoV-2 was tested negative in oropharyngeal swabs (21). Two studies have also evaluated the genomes of SARS-CoV-2 isolated from LTPPs, reporting haplotype diversity and several mutations in ORF8 (17, 22).

To explore the mechanisms underlying acute and potential biomarkers for SARS-CoV-2 infection, quantitative proteomic approaches have been adopted to assess the serum of patients with COVID-19 (23-25). Shen et al. identified some significantly differentially expressed proteins (DEPs) associated with platelet degranulation and complement system pathways in severe COVID-19 patients through sera proteomic analysis of 46 COVID-19 patients in China (23). Additionally, Gutmann et al. evaluated blood samples from hospitalized COVID-19 patients, non-COVID-19 ICU sepsis patients, and healthy controls (HCs) in the UK and identified proteins with significantly different trajectories over time. Further, 5 proteins of the complement system (C1QA, C1QB, C1QC, C4BPA, and C4BPB) and galectin-3-binding protein (LGALS3BP) were identified as interaction partners of SARS-CoV-2 spike glycoprotein (24). Similarly, Messner et al. detected 27 serum DEPs as potential biomarkers for different stages of COVID-19 patients (25).

To date, the study of molecular mechanism for LTPPs is still lacking. The quantitative proteomic approach can prove valuable for such an investigation. In this study, we performed quantitative proteomic analyses on the serum of 12 LTPPs to explore the underlying molecular mechanisms, and to identify potential therapeutic targets. The results revealed the contradictory genotypes for coagulation- and immune response related DEPs in the unity of LTPPs' bodies: in comparison with the expression of proteins in the acute infectious phase of COVID-19, some DEPs showed the same dysregulated trend in expression levels, while others showed the opposite trend. These DEPs worked together to maintain the balance of LTPPs, further endowing the contradictory steady-state with long-term persistent SARS-CoV-2 but without any clinical symptoms of COVID-19 in LTPPs.

## RESULTS AND DISCUSSION

**Clinical characteristics of 12 LTPPs.** At the end of the COVID-19 outbreak in Wuhan, 12 patients at Wuhan Pulmonary Hospital retested positive for COVID-19 and showed consistently positive RT-PCR results for SARS-CoV-2 for at least 4 weeks (29–132 days), since they had recovered from acute COVID-19 and were asymptomatic. The median carrying history was 101 days, with the longest carrying history being 132 days (Table S1). Six of the 12 LTPPs (58.33%) with at least 1 comorbidity showed longer SARS-CoV-2 carrying time, among which 5 LTPPs had hypertension and 1 had renal calculi (Table 1). During the hospitalization, in order to eliminate the long-term persistent SARS-CoV-2, antiviral treatments (Favipiravir, Qingfei Paidu decoction, etc.) were

**TABLE 1** Clinical characteristics for the 12 long-term positive patients with COVID-19 (LTPPs)

| LTPP iD | Age | Gender | Main comorbidities | Positive time | Plasma treatment |
|---|---|---|---|---|---|
| LTPP01 | 78 | F | Hypertension, diabetes | 132 | Yes |
| LTPP02 | 44 | M | Hypertension | 127 | No |
| LTPP03 | 68 | F | Hypertension, diabetes, coronary | 123 | No |
| LTPP04 | 46 | M | No | 109 | Yes |
| LTPP05 | 79 | F | Hypertension, chronic bronchitis | 109 | Yes |
| LTPP06 | 63 | M | Renal calculi | 107 | No |
| LTPP07 | 65 | M | No | 94 | No |
| LTPP08 | 37 | M | No | 94 | No |
| LTPP09 | 47 | M | Hypertension | 88 | Yes |
| LTPP10 | 69 | M | History of tuberculosis | 84 | No |
| LTPP11 | 62 | F | No | 55 | No |
| LTPP12 | 16 | M | No | 28 | No |
| Summary | 63 (46 to 68) | 8 males (66.67%) 4 female (33.33%) | 7 comorbidities (58.33%) | 101 (87–113) | 4 plasma treatment (33.33%) |

performed; thymopeptides were also used to improve the immunity of patients; four LTPPs were treated with convalescent plasma to boost their immunity. No antibiotic was used.

**Quantitative proteomic profiling showing significant differential serum protein profiles between LTPPs and health controls.** To explore the molecular mechanisms underlying the cause of disease in LTPPs, we adopted the data-independent acquisition mass spectrometry (DIA–MS) approach to perform quantitative proteomic analysis of serum samples from 12 LTPPs. Fifteen serum samples from health individuals were also included as controls (HCs) (Table S2). On average, we obtained 1,153 and 1,176 proteins from LTPP and HC samples, respectively (Fig. S1). Principal-component analysis showed clear stratification between LTPP and HC samples, indicating differential expression profiles of serum proteins between the 2 groups (Fig. S2A).

Further analysis identified 174 differently expressed proteins (DEPs) between the 2 groups (FDR ≤ 0.05; FC ≥ 2 or FC ≤ 0.5), including 146 upregulated and 28 downregulated ones (Table S3, and Fig. S2B and C), suggesting that the bodies might upregulate a majority of DEPs in serum to resist the persistent SARS-CoV-2 in the bodies of LTPPs.

**Function and pathway analyses showing significant enrichment of "coagulation" and "immune response" in the serum of LTPPs.** All DEPs were subjected to Gene Ontology (GO) (26) and Kyoto Encyclopedia of Genes and Genomes (KEGG) pathway (27) enrichment analyses (Table S4 and Table S5). In total, 383 GO biological process terms were significantly enriched (p-adjust < 0.05) (Table S4), which could be grouped into 34 parent terms (Fig. S3), and further divided into 4 classes based on their biofunctions (Fig. S4): "coagulation system," "immune response," "cell morphology and movement," and "cellular function and maintenance". Among the 4 functional classes, "coagulation system" and "immune response" showed relatively higher significance (top 1 and 2) (Fig. S4).

Further analysis revealed 17 significantly enriched pathways (p-adjust < 0.05) (Fig. S5 and Table S5), among which 3 "ones" have been reported to be involved to coagulation and immune response of SARS-CoV-2 infection ("complement and coagulation cascades," top-1; "platelet activation," top-2; "neutrophil extracellular trap formation," top-4) (28). In addition, the pathway "Coronavirus disease – COVID-19" was also enriched significantly in our study (Fig. S5 and Table S5). Interestingly, 3 bacterial infection related pathways were also significantly enriched ("*Staphylococcus aureus* infection," "shigellosis," and "bacterial invasion of epithelial cells"), implying the presence of a bacterial infection in LTPPs, despite them showing no pertinent symptoms.

**Serum protein profiles for the LTPPs without hypertension.** We then focused on 7 LTPPs without hypertension (LTPPs-NH) to eliminate the effect of the co-morbidities. PCA also showed clear stratification between LTPP-NH and HC samples (Fig. 1A). Further analysis identified 165 DEPs (FDR ≤ 0.01; FC ≥ 2 or FC ≤ 0.5) in the LTPP-NH/HC

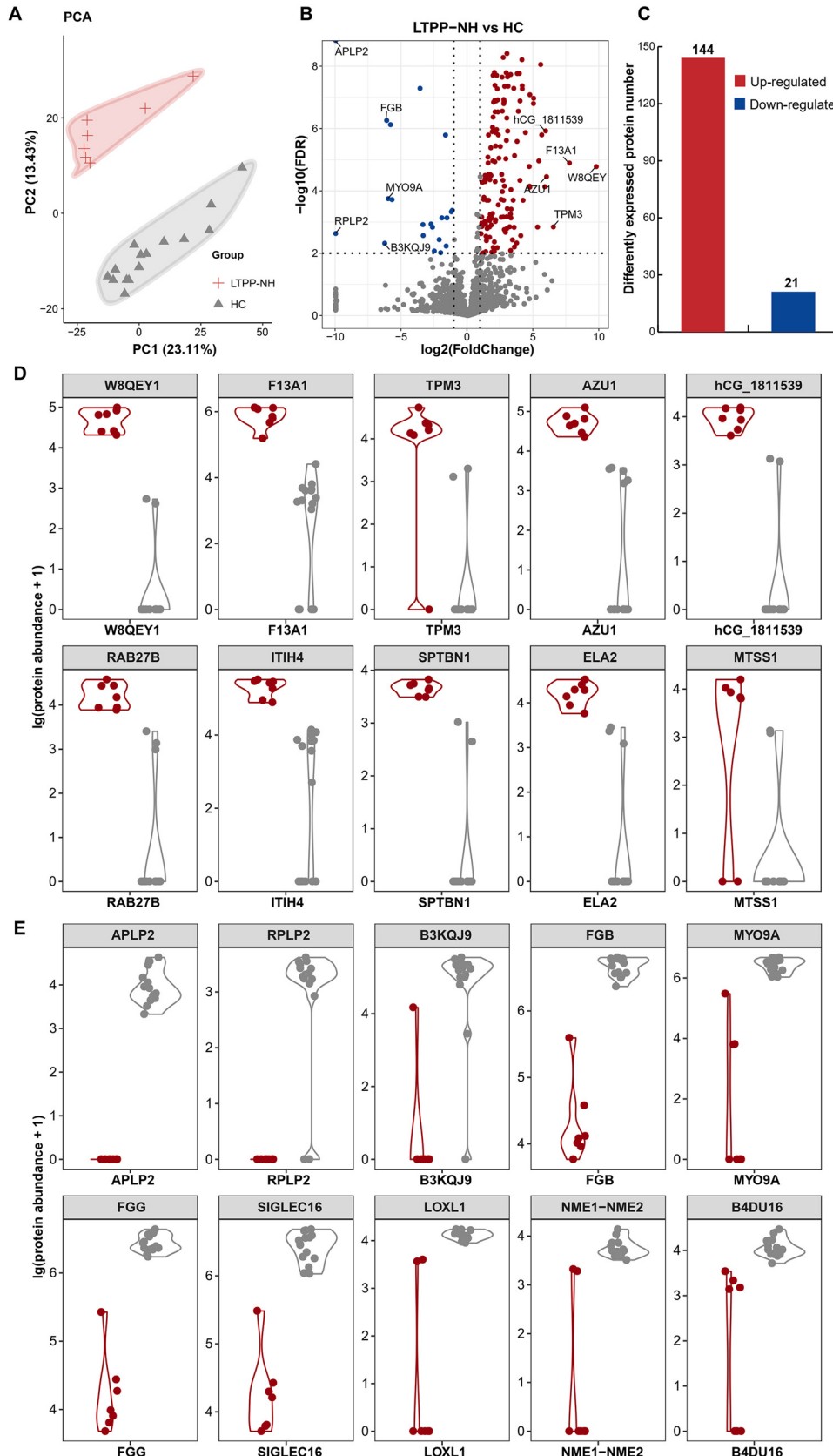

**FIG 1** The proteomic profiling of serum from LTPPs-NH and HCs. (A) Principal-component analysis of the expressed proteins between the two groups. (B) Volcano plot showing the significantly upregulated (red dots) and downregulated

group, including 144 upregulated and 21 downregulated ones (Fig. 1B and C, and Table S6), most of which (~78%) were shared with those in the LTPP/HC group, indicating smaller effect caused by co-morbidities.

We then focused on the top 10 up- and downregulated DEPs, with important roles in maintaining the contradictory balance state of LTPPs-NH (Fig. 1D and E). Among the top 10 upregulated DEPs, 1 DEP (F13A1, top 2) is related to blood coagulation, and 2 "ones" (W8QEY1 [Lactoferrin, top-1] and AZU1 [azurocidin-1, top-4]) are associated with immune response. Among the top 10 downregulated DEPs, 3 "ones" are associated with blood coagulation including APLP2 and the beta and gamma components of fibrinogen FGB and FGG.

More importantly, we found that 6 of the top 10 up- and downregulated DEPs were associated with COVID-19 (29–32). Two upregulated DEPs, AZU1 and ITIH4, displayed a consistent trend in expression as that reported by previous studies involving patients in the acute infectious phase of COVID-19 (29, 30); on the contrary, 4 DEPs (including one upregulated [F13A1] and 3 downregulated [APLP2, FGB, and FGG] DEPs) showed an opposite trend in expression (31, 32). The contradictory genotypes in the serum may be derived from the contradictory phenotype of LTPPs with long-term persistent SARS-CoV-2 but without clinical symptom, further retaining the contradictory balance state in LTPPs. Moreover, we have reason to infer that the 4 inconsistent serum proteins (F13A1, APLP2, FGB, and FGG) play roles in the fight against the long-term persistent SARS-CoV-2, further remaining no clinical symptom of COVID-19 for the LTPPs. In contrast, the 2 consistent proteins (AZU1 and ITIH4) in serum might play roles in avoiding the complete elimination of SARS-CoV-2 in the bodies of LTPPs. These proteins appear to be promising therapeutic targets for long-term persistent COVID-19, but further studies are warranted.

**Significant enrichment of "coagulation" and "immune response" in the serum of LTPPs-NH.** All the 165 DEPs of LTPPs-NH were then subjected to GO (26) and KEGG pathway (27) enrichment analyses (Table S7 and Table S8). In total, 256 GO biological process terms were significantly enriched (p-adjust < 0.05), which were classified into 28 parent terms (Fig. 2 and Fig. S6). They were further grouped into 4 same classes as those in the LTPPs based on their biofunctions (Fig. 2). Among the 4 classes in the LTPPs-NH, "coagulation system" and "immune response" ranked top-1 and top-2 (Fig. 2), which were also in agreement with those in the LTPPs. Herein, the coagulation system functional class includes 4 biological processes: blood coagulation, coagulation, wound healing, and blood coagulation, fibrin clot formation; the immune response functional class includes 7 biological processes: antimicrobial humoral response, cell chemotaxis, leukocyte migration, defense response to bacterium, regulation of endothelial cell chemotaxis, regulation of phagocytosis, and regulation of MHC class II biosynthetic process.

Further analysis revealed 12 significantly enriched pathways (p-adjust < 0.05) in the LTPPs-NH, all of which were also significantly enriched in the LTPPs, indicating small effect of comorbidity compared with SARS-CoV-2 infection (Fig. 3 and Table S8). The ranking orders of the 3 significantly enriched coagulation and immune response related pathways (complement and coagulation cascades: top-1, platelet activation: top-2, neutrophil extracellular trap formation: top-4) (28) in the LTPPs-NH were the same as those in the LTPPs (Fig. 3 and Fig. S5). In addition, the Coronavirus disease – COVID-19 pathway was also significantly enriched in the LTPPs-NH (Fig. 3 and Table S8). More interestingly, all 3 aforementioned bacterial infection related pathways (*Staphylococcus aureus* infection, shigellosis, and bacterial invasion of epithelial cells) in the LTPPs were also significantly enriched in the LTPPs-NHs.

**FIG 1** Legend (Continued)
(blue dots) proteins between the 2 groups (FC > 2). (C) Bar plot representing the number of differently expressed proteins (DEPs). (D) Box plots showing the top 10 significantly upregulated proteins in LTPPs-NH compared with HCs. (E) Box plots showing the top 10 significantly downregulated proteins in LTPPs-NH compared with HCs.

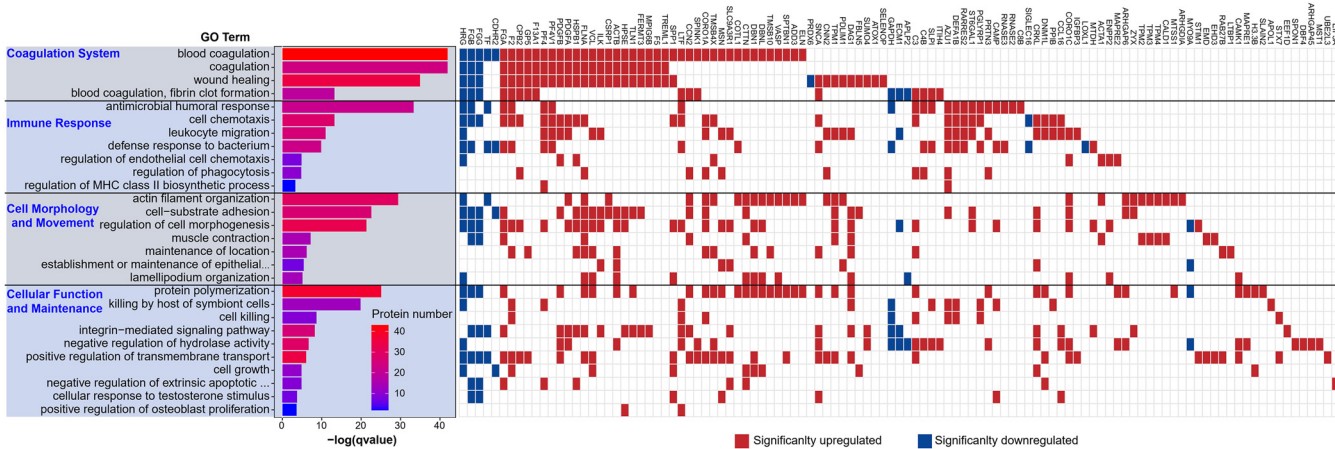

**FIG 2** The significantly enriched GO biological processes and the corresponding DEPs in the LTPPs-NH. The left panel represents four classes based on their differential GO biofunction items. The middle panel indicates the significance (adjusted *P*-value) and protein numbers of each GO item. The right panel depicts the DEPs in each GO item.

**Facilitation-inhibition balance of coagulation-related DEPs playing potential roles in maintaining the long-term positive stage for the LTPPs-NH.** In total, 63 DEPs were identified to be involved in 4 coagulation-related functions and 2 coagulation-related pathways, including 54 upregulated and 9 downregulated ones (Fig. 2 and Fig. 3). Importantly, 7 out of 63 DEPs have been reported to promote clotting, displaying the same dysregulated trend as that previously reported for patients in the acute infectious phase of COVID-19, which might be one of the possibly key reasons for the incomplete elimination of SARS-CoV-2 in LTPPs (Fig. 4 and Table 2). Specifically, 2 crucial coagulation factors (F2: thrombin, activated blood coagulation factor II; F5: coagulation factor V), 2 platelet activators (TLN1 and FERMT3), and one fibrinolysis inhibitor (CPB2), were significantly upregulated in the serum of LTPPs to facilitate blood clots (33–37). It is notable that TLN1 has been reported to be one critical hub protein and potential therapeutic target for patients with COVID-19 (36). Our research also showed significantly downregulated HRG and ECM1 proteins. HRG evidently facilitates blood

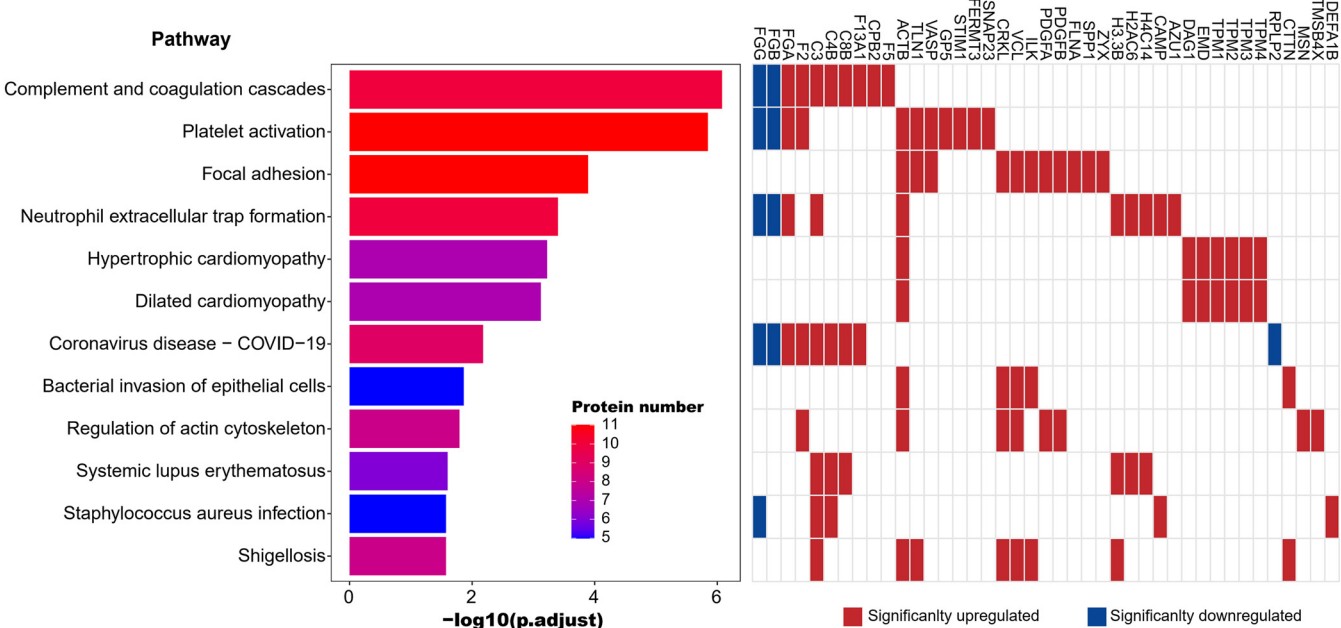

**FIG 3** The significantly enriched KEGG pathways and the corresponding DEPs in the LTPPs-NH. The left panel represents KEGG pathways. The middle panel indicates the significance (adjusted *P*-value) and protein numbers of each pathway. The right panel depicts the DEPs in each pathway.

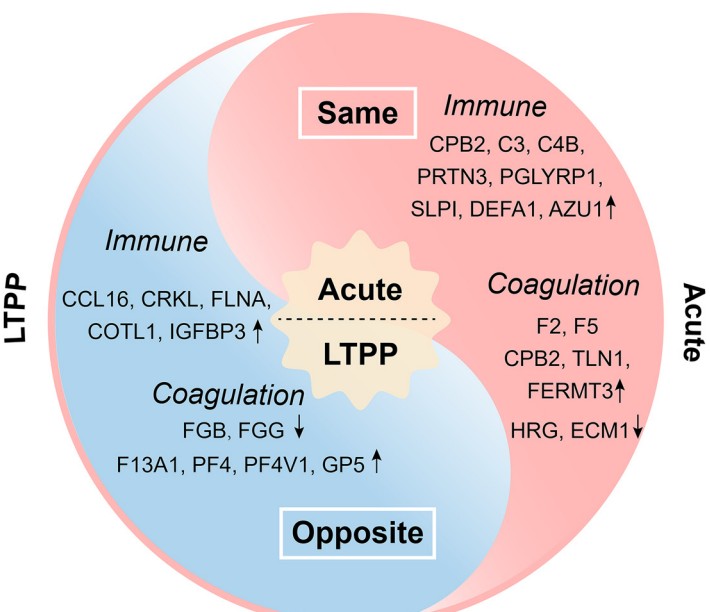

**FIG 4** The DEPs in the serums of LTPPs-NH with the same and opposite expression trends compared to those of the acute COVID-19 patients. The DEPs in the pink area represent those with the same expression trends as previously reported in the acute COVID-19 patients, including seven coagulation-related ones and 9 immune related ones. The DEPs in the blue area indicate those with the opposite expression trends compared to those in the acute COVID-19 patients, including 6 coagulation-related ones and 5 immune related DEP.

clots in patients with severe COVID-19 and is a biomarker for the poor prognosis of COVID-19 (23, 38). ECM1 has also been reported to be downregulated in COVID-19 patients due to strong association with activated partial thromboplastin time (39).

On the contrary, 6 of the 63 coagulation-related DEPs showed inhibition for blood clots in the serum of LTPPs, displaying an opposite trend to those previously reported for patients in the acute phase of COVID-19, which might be the reason for the bodies of LTPPs to retain control of clinical symptoms under long-term persistence of SARS-CoV-2 (Fig. 4 and Table 3). Herein, the beta and gamma components of fibrinogen, FGB and FGG were significantly downregulated in the serum of LTPPs to inhibit blood clots, while they were significantly upregulated in the COVID-19 and SARS patients (31, 40). In addition, the expression levels of 4 DEPs (F13A1, PF4, PF4V1, and GP5) were significantly upregulated, which is contrary to those reported for patients in the acute phase of COVID-19 (23, 31, 35, 39, 41–43). The decrease in the expression of the platelet factor PF4 is evidently associated with poor prognosis in patients with SARS (43).

In total, we identified 7 coagulation-related DEPs with the same dysregulated trend in expression as those previously reported for patients in the acute infectious phase of COVID-19, which might be caused by long-term stimulation of persistent SARS-CoV-2 infection in LTPPs, further preventing them from complete elimination. On the other hand, 6 coagulation-related DEPs showed the opposite trend in expression; these appear to potentially play a role in the fight against persistent SARS-CoV-2 infection, further retaining control of COVID-19 clinical symptoms in LTPPs (Fig. 4, Table 2, and Table 3).

**Significantly activated immune response related DEPs in the serum for fighting against long-term persistent SARS-CoV-2 in the LTPPs-NH.** A total of 62 DEPs were involved in 7 enriched biological processes and 2 enriched pathways concerning immune response in the serum of LTPPs-NH, including 53 significantly upregulated and 9 significantly downregulated ones. Most immune response related DEPs were upregulated (86.67%), indicating that the immune system of LTPPs was activated to tackle persistent SARS-CoV-2 infection. Importantly, 7 significantly upregulated DEPs

**TABLE 2** 16 DEPs with the same expression trends compared with acute stage COVID-19 patients

| Function class | Protein | Description | Name | Log2FC | FDR | Change |
|---|---|---|---|---|---|---|
| Coagulation system | P00734 | Prothrombin | F2 | 2.748696 | 1.61E-07 | Up-regulation |
| Coagulation system | A0A0A0MRJ7 | Coagulation factor V | F5 | 2.766003 | 1.43E-07 | Up-regulation |
| Coagulation system | A0A1S5UZ07 | Talin-1 | TLN1 | 2.170414 | 1.31E-07 | Up-regulation |
| Coagulation system | Q86UX7 | Fermitin family homolog 3 | FERMT3 | 1.666882 | 0.001444 | Up-regulation |
| Coagulation system | A0A087WSY5 | Carboxypeptidase B2 | CPB2 | 1.597087 | 0.000104 | Up-regulation |
| Coagulation system | P04196 | Histidine-rich glycoprotein | HRG | −1.93855 | 0.001795 | Down-regulation |
| Coagulation system | A0A140VJI7 | Testicular tissue protein Li 61 | ECM1 | −1.68704 | 5.74E-06 | Down-regulation |
| Immune response | A0A087WSY5 | Carboxypeptidase B2 | CPB2 | 1.597087 | 0.000104 | Up-regulation |
| Immune response | P01024 | Complement C3 | C3 | 3.164229 | 3.85E-05 | Up-regulation |
| Immune response | P0C0L5 | Complement C4-B | C4B | 2.576205 | 0.000166 | Up-regulation |
| Immune response | D6CHE9 | Proteinase 3 (Serine proteinase, neutrophil, Wegener granulomatosis autoantigen), isoform CRA_a | PRTN3 | 1.487113 | 0.00247 | Up-regulation |
| Immune response | O75594 | Peptidoglycan recognition protein 1 | PGLYRP1 | 1.550032 | 0.001193 | Up-regulation |
| Immune response | P03973 | Antileukoproteinase | SLPI | 2.569218 | 0.000451 | Up-regulation |
| Immune response | A0A024RDE2 | Secreted phosphoprotein 1 (Osteopontin, bone sialoprotein I, early T-lymphocyte activation 1), isoform CRA_b | SPP1 | 2.519116 | 8.34E-05 | Up-regulation |
| Immune response | P59665 | Neutrophil defensin 1 | DEFA1 | 2.435265 | 0.000109 | Up-regulation |
| Immune response | P20160 | Azurocidin | AZU1 | 5.696308 | 7.26E-07 | Up-regulation |

(CPB2, C3, C4B, PRTN3, PGLYRP1, SLP1, and SPP1) showed consistent expression trend with COVID-19 patients (39, 44–48) (Fig. 4 and Table 2). Among them, 4 "ones" (CPB2, C3, C4B, and PRTN3) have been reported to be potential biomarkers for the severity of COVID-19 (44–47): CPB2 has been reported to be a biomarker candidate for COVID-19 severity due to its differential over-expressions in various types of COVID-19 patients (44); the complement C3 has been applied to predict adverse clinical consequences of COVID-19 patients due to its negative correlation with the severity/mortality of COVID-19 (46); C4B has been documented to be a probable classification marker for predicting the severity of COVID-19 through support vector machine model (45); PRTN3 has been reported to be a potential biomarker of COVID-19 because of its specifical expression in COVID-19 patients (47). In addition, PGLYRP1 and SLPI have been reported to be gradually augmented with COVID-19 disease severity through participating neutrophil degranulation (39); SPP1 could promote SARS-CoV-2 cell entry and replication by increasing furin expression (48). Collectively, the upregulation of these proteins might be caused by long-term stimulation of persistent SARS-CoV-2 infection in LTPPs, further preventing them from complete elimination.

In contrast, 5 upregulated immune related DEPs (CCL16, CRKL, FLNA, COTL1, and IGFBP3) in the serum of LTPPs-NH displayed opposite trend compared with the acute phase of COVID-19 patients (23, 39, 47) (Fig. 4 and Table 3). It is noting that CCL16 has been reported to be associated with the recruitment of monocytes and lymphocytes during COVID-19 progression in 5 studies (49, 50), as a result, we have reason to infer that significantly upregulated CCL16 in the serum of LTPPs may play important roles to resist long-term persistent SARS-CoV-2 infection and retain control of COVID-19 clinical symptoms in LTPPs (49, 50).

Interestingly, the expression levels of 2 enzymes participating in antimicrobial immune response (DEFA1 and AZU1) (51, 52) were significantly upregulated in LTPPs-NH, suggestive of accompanying bacterial infection, which were consistent with the aforementioned enriched bacterial infection related pathways (Fig. 4 and Table 2). Similarly, previous studies have found a significant upregulation of the expression of some bacterial infection related genes in patients in the acute phase of COVID-19 (51, 52). It is noting that AZU1 is a neutrophil granule-derived antibacterial glycoprotein through monocyte- and fibroblast-specific chemotaxis (52), which has also been

**TABLE 3** 11 DEPs with opposite expression trends as those in acute stage COVID-19 patients

| Function class | Protein | Description | Name | Log2FC | FDR | Change |
|---|---|---|---|---|---|---|
| Coagulation system | P02675 | Fibrinogen beta chain | FGB | −5.94858 | 2.43E-06 | Down-regulation |
| Coagulation system | P02679 | Fibrinogen gamma chain | FGG | −5.69905 | 3.07E-06 | Down-regulation |
| Coagulation system | P02776 | Platelet factor 4 | PF4 | 1.887417 | 4.58E-05 | Up-regulation |
| Coagulation system | P10720 | Platelet factor 4 variant | PF4V1 | 2.219829 | 3.66E-06 | Up-regulation |
| Coagulation system | P40197 | Platelet glycoprotein V | GP5 | 3.607381 | 1.48E-05 | Up-regulation |
| Coagulation system | P00488 | Coagulation factor XIII A chain | F13A1 | 7.729495 | 1.29E-05 | Up-regulation |
| Immune response | O15467 | C-C motif chemokine 16 | CCL16 | 1.160563 | 0.000706 | Up-regulation |
| Immune response | P46109 | Crk-like protein | CRKL | 1.301672 | 0.002281 | Up-regulation |
| Immune response | P21333 | Filamin-A | FLNA | 2.122492 | 1.27E-09 | Up-regulation |
| Immune response | A0A384MTY2 | Epididymis secretory sperm binding protein | COTL1 | 1.617894 | 0.000401 | Up-regulation |
| Immune response | A6XND0 | Insulin-like growth factor-binding protein 3 | IGFBP3 | 2.720378 | 1.27E-08 | Up-regulation |

reported to be a potential therapeutic target for COVID-19 due to its high expression in the acute phase of COVID-19 patients (29).

**Conclusions.** We reported a new type of COVID-19 patients, who were retested positive after hospital discharge, with long-term persistent SARS-CoV-2 but without COVID-19 clinical symptom in their bodies. The underlying mechanism of the contradictory phenomenon were further explored by quantitative proteomics on the serum of LTPPs, and revealed a unity of contradictory proteomic genotypes for the coagulation- and immune response related DEPs in the bodies of LTPPs, which ranked the 1st and 2nd among all the DEPs and the significantly enriched functions and pathways. We also reanalyzed the proteomic profiles of the LTPPs without hypertension (LTPPs-NH), and achieved similar results from DEP, GO function, and KEGG pathway analyses, demonstrating small effects caused by co-morbidities compared with SARS-CoV-2 infection. Herein, 16 DEPs (Fig. 4 and Table 2) showed the same dysregulated trend in expression as that previously reported for patients in the acute phase of COVID-19, which might be caused by long-term stimulation of persistent SARS-CoV-2 infection in LTPPs, preventing the complete elimination of SARS-CoV-2. In contrast, 11 DEPs (Fig. 4 and Table 3) showed the opposite trend in expression; these seem to play a key role in conferring resistance to long-term persistent SARS-CoV-2 infection to retain control of COVID-19 clinical symptoms in LTPPs. These contradictory expression trends seem to work in harmony to maintain the balance of LTPPs, further endowing the contradictory steady-state with long-term persistent SARS-CoV-2 but without clinical symptoms in LTPPs.

This study has some limitations. It should be noted that the detection of nucleic acids after recovery does not indicate transmissible SARS-CoV-2 (53). In addition, considering the serious threat posed by the COVID-19 pandemic and the burden on clinicians in Wuhan during March to June 2020, it was challenging to obtain more samples for further analyses. In addition, as we are not certified to handle SARS-CoV-2, pertinent experiments could not be performed to verify our results; we only obtained the longitudinal RT-PCR testing results of the patients (Table S1) rather than the Ct values for the RT-PCR results. Nonetheless, we believe that the aforementioned DEPs can serve as promising therapeutic targets for COVID-19, though further studies are warranted.

## MATERIALS AND METHODS

**Protein preparation.** Serum samples were obtained from LTPPs by clotting blood at room temperature, followed by centrifugation at 1,000 × $g$ for 3 min at 4°C. High-abundance proteins were depleted by Pierce Top 12 (Thermo Scientific) protein depletion columns, and total protein concentration was measured using the Bradford protein assay kit (Bio-Rad). Protein digestion was performed as previously described (54–56): DB lysis buffer (8 M urea, 100 mM TEAB, pH 8.5), trypsin, and 100 mM TEAB buffer were added to protein samples, followed by incubation at 37°C for 4 h. The mixture was subsequently digested with trypsin (Promega) in the presence of $CaCl_2$ at 37°C overnight, followed by desalination on a C18 cartridge. The samples were then lyophilized for liquid chromatography (LC)–MS/MS analysis.

**LC-MS/MS.** Samples were quantified using data-independent acquisition (DIA) mode by Q Exactive HF-X mass spectrometer (Thermo Scientific) coupled with an Ultimate 3000 UHPLC LC system (Thermo Scientific). Lyophilized peptides were first dissolved in solution A (100% water and 0.1% formic acid),

and separated on an analytical column (C18, Thermo Scientific) with a gradient of 8% to 95% solution B (80% acetonitrile and 0.1% formic acid) for 80 min at a flow rate of 600 nL min$^{-1}$. Peptides eluted from the microcapillary column were directly electrosprayed into a mass spectrometer using a Nanospray Flex (ESI) ion source; the spray voltage was 2.4 kV and ion-transport capillary temperature was 320°C. In terms of HF-X settings, the mass range of MS1 was set at 400 to 1250 $m/z$, with the resolution being 120,000. The automatic gain control was set to 3,000,000, and the maximum ion injection time to 50 ms. In terms of DIA settings, the mass range of MS2 was 400 to 1250 $m/z$, and MS2 scans were performed at a resolution of 30,000, with the automatic gain control set to 1,000,000, automatic maximum ion injection time, and normalized collision energy of 27%.

**Proteins identification and quantitation.** The spectra from each fraction were searched against the UniProt database using Proteome Discoverer 2.2 (Thermo). The parameters were as follows: mass tolerance for precursor ion, 10 ppm; mass tolerance for product ion, 0.02 Da; fixed modification, carbamidomethyl (C); dynamic modification, oxidation methionine (M); N-terminal modification, acetylation; and maximum missed cleavages, 2. High-quality search results were defined as follows: peptide spectrum matches with a credibility of >99%; proteins with at least one unique peptide; and FDR ≤ 0.01.

Next, the search results obtained by Proteome Discoverer 2.2 were imported into Spectronaut v14.0 (Biognosys) to generate a library. All eligible peptides and ions were selected from the spectra, as previously described (57), which led to the generation of a "Target List". DIA data were also imported to calculate the quantity of peptides according to the "Target List".

**Identification of DEPs.** The proteins were statistically analyzed by $t$ test. The following criteria were applied to identify DEPs of LTPPs: FDR < 0.05 and fold change ≥ 2 or ≤ 0.5. The criteria of the DEPs of LTPPs-NH were FDR < 0.01 and fold change ≥ 2 or ≤ 0.5. All DEPs with significant changes were subjected to GO and KEGG pathway enrichment analyses.

**Ethical approval.** This study was approved by the Medical Ethical Committee of Wuhan Pulmonary Hospital and the Ethical Committee of Beijing Institute of Genomics, Chinese Academy of Sciences, China National Center for Bioinformation.

**Data availability.** The mass spectrometry proteomics data and corresponding metadata have been deposited to the ProteomeXchange Consortium via the PRIDE (58) partner repository with the data set identifier PXD036609.

## SUPPLEMENTAL MATERIAL

Supplemental material is available online only.
**SUPPLEMENTAL FILE 1**, XLSX file, 0.2 MB.
**SUPPLEMENTAL FILE 2**, PDF file, 1 MB.

## ACKNOWLEDGMENTS

We thank Peng Peng, Guan Liu, Li Li, Yingxia He, and Qi Zhu in Wuhan Pulmonary Hospital for their support in collecting samples and providing clinical information about the patients. We also thank the patients for their willingness to participate in this study.

This work was supported by Beijing Natural Science Foundation (Grant No. M21009) and Funds for International Cooperation and Exchange of the National Natural Science Foundation of China (Grant No. 32061143024).

F.C. conceived the study. C.L., Jie. W., M.C., and H.L. performed the bioinformatics analyses. L.Y., Y.J., T.L., S.L., Jing. W., X.H., B.T., H.W., and W.Z. carried out the experimental analyses. C.L., S.X., X.H., B.T., H.W., and C.J. drew the figures. F.C., C.L., and L.Y. wrote the manuscript. All authors read and approved the final manuscript.

We declare no competing interests.

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
