## [Reviewer comments · Microbiology Spectrum]

Microbiology Spectrum

Serum proteomic analysis for a new type of long-term persistent COVID-19 patients in Wuhan

Cuidan Li, Liya Yue, Yingjiao Ju, Jie Wang, Mengfan Chen, Hao Lu, Sitong Liu, Tao Liu, Jing Wang, Xin Hu, Bahetibieke Tuohetaerbaike, Hao Wen, Wenbao Zhang, Sihong Xu, Chunlai Jiang, and Fei Chen

Corresponding Author(s): Fei Chen, Beijing Institute of Genomics, Chinese Academy of Sciences

Review Timeline:

Submission Date:	April 5, 2022
Editorial Decision:	June 12, 2022
Revision Received:	August 2, 2022
Editorial Decision:	September 6, 2022
Revision Received:	September 13, 2022
Accepted:	October 7, 2022

Editor: Rafael A. Medina

Reviewer(s): Disclosure of reviewer identity is with reference to reviewer comments included in decision letter(s). The following individuals involved in review of your submission have agreed to reveal their identity: Marcelo Alarcon (Reviewer #1)

Transaction Report:

DOI: <https://doi.org/10.1128/spectrum.01270-22>

June 12, 2022

Prof. Fei Chen
Beijing Institute of Genomics, Chinese Academy of Sciences
Beijing 100101
China

Re: Spectrum01270-22 (Serum proteomic analysis for a new type of long-term persistent COVID-19 patients in Wuhan)

Dear Prof. Fei Chen:

Thank you for submitting your manuscript to Microbiology Spectrum. You will see from the referees' comments that additional information needs to be provided. Please update the introduction to provide a better context to the study. Include information on viral kinetics of the studied individuals (address the possibility of false positives), and the potential role of comorbidities in coagulation and immune responses. Please also revise and provide further details on the statistical analyses, and address the comment from reviewer 2 on the potential bias of DIA. We ask that this be provided, before we consider your manuscript further.

Link Not Available

Sincerely,

Rafael A. Medina

Journals Department
Reviewer comments:

Reviewer #1 (Comments for the Author):

,

Reviewer #2 (Comments for the Author):

This study examines a small set of patients that test positive for SARS-CoV-2 by RT-PCR while not displaying symptoms. The

authors term these patients long term persistent COVID-19 patients. The relationship between SARS-CoV-2 infection and chronic symptoms (or lack thereof) is important to dissect, this study needs additional analysis to confirm a relationship between the proteomic changes and infection.

The authors define this LTPP group, but do not show the RT-PCR testing results. From the authors own citations, some of these results could be false positives. Without information on the viral kinetics of it is difficult to see variability and relationships between patients. What is the relationship between the proteomics and the kinetics?

Additionally a number of patients had significant co-morbidities such as hypertension and diabetes, the authors should examine whether changes in coagulation and immune response are related to these co-morbidities rather than infection.

Similarly information should be given on the healthy controls, it is impossible to determine if they are age-matched or matched for co-morbidities. If not matched for age, or co-morbidities this could significantly skew the results.

The statistical analyses for the proteomics seem simplified-just p values are given, no FDR or corrections for multiple testing. Especially with the small N, the authors should reanalyze the data and see if the results hold true.

DIA tends to bias to abundant proteins. Coagulation defects have been well-described in hospitalized COVID-19 patients, the authors don't have any data to suggest this pathway is involved in persistence and should soften their language.

No information is given on treatments other than plasma treatment, this should be included

Staff Comments:

Preparing Revision Guidelines

Please return the manuscript within 60 days; if you cannot complete the modification within this time period, please contact me. If you do not wish to modify the manuscript and prefer to submit it to another journal, please notify me of your decision immediately so that the manuscript may be formally withdrawn from consideration by Microbiology Spectrum.

Response to reviewers:

This study examines a small set of patients that test positive for SARS-CoV-2 by RT-PCR while not displaying symptoms. The authors term these patients long term persistent COVID-19 patients. The relationship between SARS-CoV-2 infection and chronic symptoms (or lack thereof) is important to dissect, this study needs additional analysis to confirm a relationship between the proteomic changes and infection.

1. The authors define this LTPP group, but do not show the RT-PCR testing results. From the authors own citations, some of these results could be false positives. Without information on the viral kinetics of it is difficult to see variability and relationships between patients. What is the relationship between the proteomics and the kinetics?

Answer: Many thanks for pointing out this. As per your suggestion, we have provided the detailed information for the longitudinal RT-PCR testing results of the patients (see figure and table below). All the patients were tested positive for a long time, indicating long-term persistent of SARS-CoV-2 rather than false positives for the patients. We have also added a new supplementary table of the RT-PCR results to the revised MS (Table S1). We also tried our best to search for the Ct values of these RT-PCR results in the medical record system of the hospital, but failed. As a results, we have added the limitation in the revised MS (Page 14, Line 327-328).

Figure R1.

Figure R1. Clinical courses of the 12 LTPPs. The time lines show the timepoints of symptom onset, hospital readmission, and discharge of each case.

Table R1. The longitudinal RT-PCR testing results for the LTPP01*

Patient ID	Sex	Hospital numbers	Report Time	Targeting	RT-PCR Results
LTPP01	Female	3972340	2020/03/31 19:52:46	N region	Positive
LTPP01	Female	3972340	2020/03/31 19:52:46	ORF1ab	Positive
LTPP01	Female	3972340	2020/04/01 15:24:45	N region	Negative
LTPP01	Female	3972340	2020/04/01 15:24:45	ORF1ab	Negative
LTPP01	Female	3972340	2020/04/01 18:58:03	N region	Positive
LTPP01	Female	3972340	2020/04/01 18:58:03	ORF1ab	Positive
LTPP01	Female	3972340	2020/04/01 18:58:35	N region	Positive
LTPP01	Female	3972340	2020/04/01 18:58:35	ORF1ab	Positive
LTPP01	Female	3972340	2020/04/02 14:14:38	N region	Weak positive
LTPP01	Female	3972340	2020/04/02 14:14:38	ORF1ab	Positive
LTPP01	Female	3972340	2020/04/03 09:18:37	N region	Negative
LTPP01	Female	3972340	2020/04/03 09:18:37	ORF1ab	Negative
LTPP01	Female	3972340	2020/04/03 09:49:53	N region	Negative
LTPP01	Female	3972340	2020/04/03 09:49:53	ORF1ab	Negative
LTPP01	Female	3972340	2020/04/05 09:07:57	N region	Negative
LTPP01	Female	3972340	2020/04/05 09:07:57	ORF1ab	Negative
LTPP01	Female	3972340	2020/04/05 15:41:14	N region	Positive
LTPP01	Female	3972340	2020/04/05 15:41:14	ORF1ab	Positive
LTPP01	Female	3972340	2020/04/05 15:51:13	N region	Weak positive
LTPP01	Female	3972340	2020/04/05 15:51:13	ORF1ab	Weak positive
LTPP01	Female	3972340	2020/04/05 15:51:49	N region	Positive
LTPP01	Female	3972340	2020/04/05 15:51:49	ORF1ab	Positive
LTPP01	Female	3972340	2020/04/07 09:09:49	N region	Negative
LTPP01	Female	3972340	2020/04/07 09:09:49	ORF1ab	Negative
LTPP01	Female	3972340	2020/04/07 10:02:38	N region	Negative
LTPP01	Female	3972340	2020/04/07 10:02:38	ORF1ab	Negative
LTPP01	Female	3972340	2020/04/07 14:17:27	N region	Positive
LTPP01	Female	3972340	2020/04/07 14:17:27	ORF1ab	Positive
LTPP01	Female	3972340	2020/04/07 14:19:46	N region	Weak positive
LTPP01	Female	3972340	2020/04/07 14:19:46	ORF1ab	Weak positive
LTPP01	Female	3972340	2020/04/10 09:19:58	N region	Negative
LTPP01	Female	3972340	2020/04/10 09:19:58	ORF1ab	Negative
LTPP01	Female	3972340	2020/04/10 10:13:10	N region	Negative
LTPP01	Female	3972340	2020/04/10 10:13:10	ORF1ab	Negative
LTPP01	Female	3972340	2020/04/10 19:31:55	N region	Positive
LTPP01	Female	3972340	2020/04/10 19:31:55	ORF1ab	Positive
LTPP01	Female	3972340	2020/04/10 19:34:19	N region	Positive
LTPP01	Female	3972340	2020/04/10 19:34:19	ORF1ab	Positive
LTPP01	Female	3972340	2020/04/13 09:14:50	N region	Negative
LTPP01	Female	3972340	2020/04/13 09:14:50	ORF1ab	Negative
LTPP01	Female	3972340	2020/04/13 14:56:33	N region	Positive

LTPP01	Female	3972340	2020/04/13 14:56:33	ORF1ab	Positive
LTPP01	Female	3972340	2020/04/13 14:57:41	N region	Positive
LTPP01	Female	3972340	2020/04/13 14:57:41	ORF1ab	Positive
LTPP01	Female	3972340	2020/04/13 19:29:04	N region	Positive
LTPP01	Female	3972340	2020/04/13 19:29:04	ORF1ab	Positive
LTPP01	Female	3972340	2020/04/18 15:35:25	N region	Negative
LTPP01	Female	3972340	2020/04/18 15:35:25	ORF1ab	Negative
LTPP01	Female	3972340	2020/04/19 14:28:14	N region	Positive
LTPP01	Female	3972340	2020/04/19 14:28:14	ORF1ab	Positive
LTPP01	Female	3972340	2020/04/23 09:54:09	N region	Negative
LTPP01	Female	3972340	2020/04/23 09:54:09	ORF1ab	Negative
LTPP01	Female	3972340	2020/04/23 14:20:45	N region	Positive
LTPP01	Female	3972340	2020/04/23 14:20:45	ORF1ab	Positive
LTPP01	Female	3972340	2020/04/27 17:57:15	N region	Positive
LTPP01	Female	3972340	2020/04/27 17:57:15	ORF1ab	Positive
LTPP01	Female	3972340	2020/05/02 15:03:50	N region	Negative
LTPP01	Female	3972340	2020/05/02 15:03:50	ORF1ab	Negative
LTPP01	Female	3972340	2020/05/02 18:08:14	N region	Positive
LTPP01	Female	3972340	2020/05/02 18:08:14	ORF1ab	Positive
LTPP01	Female	3972340	2020/05/08 15:01:08	N region	Negative
LTPP01	Female	3972340	2020/05/08 15:01:08	ORF1ab	Negative
LTPP01	Female	3972340	2020/05/08 18:43:54	N region	Positive
LTPP01	Female	3972340	2020/05/08 18:43:54	ORF1ab	Positive
LTPP01	Female	3972340	2020/05/14 14:25:16	N region	Positive
LTPP01	Female	3972340	2020/05/14 14:25:16	ORF1ab	Positive
LTPP01	Female	3972340	2020/05/20 19:17:46	N region	Positive
LTPP01	Female	3972340	2020/05/20 19:17:46	ORF1ab	Positive
LTPP01	Female	3972340	2020/05/24 18:52:36	N region	Positive
LTPP01	Female	3972340	2020/05/24 18:52:36	ORF1ab	Positive
LTPP01	Female	3972340	2020/05/27 14:40:29	N region	Positive
LTPP01	Female	3972340	2020/05/27 14:40:29	ORF1ab	Positive
LTPP01	Female	3972340	2020/05/31 09:12:30	N region	Negative
LTPP01	Female	3972340	2020/05/31 09:12:30	ORF1ab	Negative
LTPP01	Female	3972340	2020/06/04 10:00:24	N region	Negative
LTPP01	Female	3972340	2020/06/04 10:00:24	ORF1ab	Negative
LTPP01	Female	3972340	2020/06/06 14:49:52	N region	Negative
LTPP01	Female	3972340	2020/06/06 14:49:52	ORF1ab	Negative

* Here, we only presented the longitudinal RT-PCR testing results for LTPP01, and the information for all the 12 LTPPs were provided as the supplementary table SXXX.

2. Additionally a number of patients had significant co-morbidities such as hypertension and diabetes, the authors should examine whether changes in coagulation and immune response are related to these co-morbidities rather than infection.

Answer: Many thanks for your comment. According to your suggestion, we have carefully reanalyzed the proteomic profiles of seven LTPPs without co-morbidities such as hypertension, diabetes, and coronary (termed as LTPPs-NH). The results demonstrated that the changes in coagulation and immune response were indeed derived from the long-term infection of SARS-CoV-2 rather than the co-morbidities, due to similar results from DEP, GO function and KEGG pathway analysis between LTPP/HC and LTPPs-NH/HC groups (Page 8, Line 167-169; Page 9, Line 199-201; Page 9, Line 208-210). Herein, the “coagulation system” and “immune response” were also the top two significantly enriched functions and pathways in the LTPPs-NH/HC group, indicating smaller effect caused by co-morbidities compared with SARS-CoV-2 infection. We have also added the related results and description about LTPPs-NH in the revised MS (Page 7-12, Line 162-290).

3. Similarly information should be given on the healthy controls, it is impossible to determine if they are age-matched or matched for co-morbidities. If not matched for age, or co-morbidities this could significantly skew the results.

Answer: Many thanks for your thoughtful comment. As mentioned above, we obtained the similar results in the LTPP-NH/HC group compared with those in the LTPP/HC group, and have added related results and descriptions of the LTPPs without comorbidities (LTPPs-NH) to match for co-morbidities in the revised MS (Page 7-12, Line 162-290). In addition, according to your suggestion, we have included the information of healthy controls as a new supplementary table (Table S2). The abundance results showed no significant difference between the patients in different age groups (see figure below).

Figure R2.

Figure R2. The distribution of correlation values from patients in the same age group and those in different age groups.

4. The statistical analyses for the proteomics seem simplified-just p values are given, no FDR or corrections for multiple testing. Especially with the small N, the authors should reanalyze the data and see if the results hold true.

Answer: Many thanks for pointing out this, and we agree with you about this point. As per the suggestion, we performed FDR corrections of multiple testing for the DEPs. We obtained similar conclusions based on new DEPs, and the results hold true. We have added related results and descriptions in the revised MS (Page 6-7, Line 135-160).

5. DIA tends to bias to abundant proteins.

Answer: Many thanks for your comment, and we agree with you. In our study, we first removed the high abundant proteins by Pierce™ Top 12 (Thermo Scientific) protein depletion columns in the DIA mode to avoid bias, and have also added the detailed information in the “Methods and Materials” section (Page 14, Line 335-337).

6. Coagulation defects have been well-described in hospitalized COVID-19 patients, the authors don't have any data to suggest this pathway is involved in persistence and should soften their language.

Answer: Many thanks for your thoughtful comment. According to your suggestion, we have carefully revised the description about coagulation defects in the revised MS (Page 10, Line 220-228).

7. No information is given on treatments other than plasma treatment, this should be included.

Answer: Thanks for pointing out this. As you suggested, we have added the information about treatments in the revised MS (Page 6, Line 119-123).

September 6, 2022

Prof. Fei Chen
Beijing Institute of Genomics, Chinese Academy of Sciences
Beijing 100101
China

Re: Spectrum01270-22R1 (Serum proteomic analysis for a new type of long-term persistent COVID-19 patients in Wuhan)

Dear Prof. Fei Chen:

We appreciate your consideration of the reviewers' comments and for the submission of a revised version. Please address the following issues raised by reviewers before we can further consider your manuscript. Indicate if the proteomics dataset and metadata have been deposited in a database (e.g. such as PRIDE) and include the accession number/s in the methodology section. Please also address the caveat mentioned by the reviewer #2 in regards to the detection of nucleic acids after recovery.

Link Not Available

Sincerely,

Rafael A. Medina

Journals Department
Reviewer comments:

Reviewer #1 (Comments for the Author):

.

Reviewer #2 (Comments for the Author):

The manuscript is much approved with additional statistical testing. The authors should include a caveat in their limitations that the detection of nucleic acids after recovery does not indicate transmissible virus see:Transfusion

Staff Comments:

Preparing Revision Guidelines

Please return the manuscript within 60 days; if you cannot complete the modification within this time period, please contact me. If you do not wish to modify the manuscript and prefer to submit it to another journal, please notify me of your decision immediately so that the manuscript may be formally withdrawn from consideration by Microbiology Spectrum.

Response to reviewers:

1. The manuscript is much approved with additional statistical testing.

Answer: We appreciate your positive comments and valuable suggestions on improving the quality of our manuscript.

2. The authors should include a caveat in their limitations that the detection of nucleic acids after recovery does not indicate transmissible virus see:Transfusion. 2020 Dec;60(12):2962-2968.

Answer: Many thanks for pointing out this, and we agree with you about this point. According to your suggestion, we have added the caveat and corresponding reference to the limitation part in Line 323-324 (Page 14) as “It should be noted that the detection of nucleic acids after recovery does not indicate transmissible SARS-CoV-2 (53).” in the revised MS.

53. Ikegami S, Benirschke R, Flanagan T, Tanna N, Klein T, Elue R, Deboz P, Mallek J, Wright G, Guariglia P, Kang J, Gniadek TJ. 2020. Persistence of SARS-CoV-2 nasopharyngeal swab PCR positivity in COVID-19 convalescent plasma donors. Transfusion 60:2962-2968.

October 7, 2022

Prof. Fei Chen
Beijing Institute of Genomics, Chinese Academy of Sciences
Beijing 100101
China

Re: Spectrum01270-22R2 (Serum proteomic analysis for a new type of long-term persistent COVID-19 patients in Wuhan)

Dear Prof. Fei Chen:

I am pleased to inform you that your manuscript has been accepted, and I am forwarding it to the ASM Journals Department for publication. You will be notified when your proofs are ready to be viewed.

Sincerely,

Rafael A. Medina
Editor, Microbiology Spectrum

Journals Department
Supplemental file 8: Accept
Supplemental file 6: Accept
Supplemental file 1: Accept
Supplemental file 7: Accept
Supplemental file 5: Accept
Supplemental file 9: Accept
Supplemental file 4: Accept
Supplemental file 3: Accept
Supplemental file 2: Accept